# A Novel Approach of a Low-Cost Voltage Fault Injection Method for Resource-Constrained IoT Devices: Design and Analysis

**DOI:** 10.3390/s23167180

**Published:** 2023-08-15

**Authors:** Nicolás Ruminot, Claudio Estevez, Samuel Montejo-Sánchez

**Affiliations:** 1Department of Electrical Engineering, Universidad de Chile, Santiago 8940577, Chile; nicolas.ruminot@ug.uchile.cl (N.R.); cestevez@uchile.cl (C.E.); 2Programa Institucional de Fomento a la Investigación, Desarrollo e Innovación, Universidad Tecnológica Metropolitana, Santiago 8940577, Chile

**Keywords:** cybersecurity, fault injection attack, Internet of Things (IoT), physical attack, resource-constrained device

## Abstract

The rapid development of the Internet of Things (IoT) has brought about the processing and storage of sensitive information on resource-constrained devices, which are susceptible to various hardware attacks. Fault injection attacks (FIAs) stand out as one of the most widespread. Particularly, voltage-based FIAs (V-FIAs) have gained popularity due to their non-invasive nature and high effectiveness in inducing faults by pushing the IoT hardware to its operational limits. Improving the security of devices and gaining a comprehensive understanding of their vulnerabilities is of utmost importance. In this study, we present a novel fault injection method and employ it to target an 8-bit AVR microcontroller. We identify the optimal attack parameters by analyzing the detected failures and their trends. A case study is conducted to validate the efficacy of this new method in a more realistic scenario, focusing on a simple authentication method using the determined optimal parameters. This analysis not only demonstrates the feasibility of the V-FIA but also elucidates the primary characteristics of the resulting failures and their propagation in resource-constrained devices. Additionally, we devise a hardware/software countermeasure that can be integrated into any resource-constrained device to thwart such attacks in IoT scenarios.

## 1. Introduction

Information security seeks to protect the confidentiality, integrity, and availability of information [1]. However, this approach is insufficient to cover rapid technological developments and new security threats, giving space to new models such as the reference model of information assurance and security [2], which has a much broader scope (as explained in [3]), an ISA/IEC 62433 standard oriented to industrial cybersecurity [4], consensus-based automation, and control system cybersecurity standards.

Nowadays, security is one of the most relevant topics in embedded systems and Internet of Things (IoT) technologies. By 2025, it is expected that there will be more than 30 billion IoT connections, at almost four IoT devices per person on average [5]. IoT technologies use multiple resource-constrained devices that have a fundamental role in how humans relate to technology. Therefore, it is essential to know their vulnerabilities and weaknesses to establish clear guidelines on how to use them and what precautions users should take. The IoT will be present with increasing force in critical infrastructures, such as electrical networks, hospitals, transport, and telecommunications. The IoT will also be a fundamental part of everyday life, so information security studies regarding these technologies must be a current priority. The most common way to protect information is through encryption algorithms. Even though in resource-constrained devices, it is more complex to implement encryption, as discussed in [6,7], various options can help choose an appropriate encryption for each device, as shown in [8]. On the other hand, the branch of hardware security does not have such a clear option to protect information. Multiple hardware attacks are aimed at exploiting vulnerabilities in both access encryption and authentication methods. Therefore, hardware security studies must be a current priority.

There is a wide range of vulnerabilities in hardware security, among the most popular and dangerous are side channel analyses (SCAs), presented first in [9], and fault injection attacks (FIAs), presented for the first time in [10]. The main goal of an SCA is to break cryptographic keys using physical information leaks. On the other hand, an FIA seeks to cause a malfunction in the attacked device by exposing it to conditions outside the limits specified by the manufacturer. This can result in bypassing authentication, accessing unauthorized code, changing a logical value, and turning off or restarting a device [10].

There are several ways to produce a malfunction in devices, but in this investigation, we will focus on voltage-based FIAs (V-FIAs). A simple circuit to inject voltage faults through a crowbar circuit was presented in 2016 [11]. The authors showed that very few faults could be injected into the test devices. However, a rigorous study concerning the number and type of resulting faults was not carried out. In [12], a thorough study of clock FIA performance was carried out, where the result of skipping instructions was analyzed in detail. In 2019, the relevant impact of the glitch pulse shape on the V-FIA performance was shown. Thus, some specific glitch pulse shapes were analyzed and proposed. The shape of these pulses was determined by genetic algorithms and implemented with appropriate electronics for this purpose [13]. Controlling the shape of the pulse to optimize an attack is computationally expensive and the complexity of the attack is also high. In a V-FIA, it is assumed that the attacker is equipped with a device to execute the attack. However, an alternative approach was introduced in [14], where a model named a field-replaceable unit was proposed. This model entails modifying specific components of a device, such as the battery, screen, or sensors, during the replacement process. In the comprehensive review presented in [15], an examination of FIAs was conducted, consolidating the literature up to 2022. This review encompasses solutions, the limitations of these solutions, and challenges associated with detecting such attacks. Anomaly identification in system operations stands as one of the most widely employed approaches for devising countermeasures. Numerous endeavors have been undertaken in this realm, exploring diverse methodologies such as bio-inspired algorithms [16], multi-agent algorithms [17], and machine learning techniques [18] to detect anomalies in data streams. Nevertheless, these approaches primarily focus on collective detection among groups of devices, lacking individual protection for each device. Considering the unique characteristics of V-FIAs, it becomes imperative for each device to possess the capability of autonomously detecting anomalies in its behavior. In [19], a fault-tolerant architecture focused on the simultaneous application of proactive and reactive policies was introduced. This architecture covered five relevant phases of fault: forecasting, prevention, detection, isolation, and recovery. Two years later, the same authors presented an extension of this hybrid architecture of fault tolerance in the second layer of the IoT. In the new version [20], the authors added a fault diagnosis phase and considered new methods of fault detection and new fault recovery strategies. The integration of these constituent phases forms the foundation of the fault tolerance capability, and they hold the potential to serve as a valuable reference for developing effective countermeasures in diverse scenarios. Furthermore, the economic viability of FIA techniques was analyzed in [21], providing a comparative assessment of the costs associated with each attack. An example of an FIA’s scope is the Starlink hack presented at the BlackHat 2022 conference. The FIA allowed bypassing the secure boot protections from the user terminal by tampering with the electrical power rails at just the right time during boot-up [22].

In this paper, we present a novel V-FIA approach for resource-constrained devices, which can be implemented by generating a pulse with a semi-fixed shape and using affordable equipment. This method provides more control flexibility than the technique described in [11]. Furthermore, the control of the shape of the pulse is not as complex as that proposed in [13] and can be carried out with inexpensive equipment. Finally, in comparison with the V-FIAs in [21], our proposal can be carried out with very low-cost equipment and can still be used to bypass access methods on AVR devices. The behavior of the generated faults was investigated through various tests to understand this phenomenon. In this study, the optimal parameters for generating exploitable failures were determined with the new method in an 8-bit AVR microcontroller, and the results were presented to illustrate the fault trends. In addition, a passive countermeasure designed to prevent an adversary from finding the optimal parameters to attack is proposed. This countermeasure, unlike those presented in [16,17,18], has an anomaly detection mechanism that can be implemented on any resource-constrained device. Furthermore, it does not increase the possibility of causing an information leak through side channel in the operations’ redundancy countermeasures, as do those presented in [23,24]. The main contributions of this article are outlined next.
A new approach to control the outcome of a fault injection attack through the monitoring residual voltage in the voltage interruption window.A study of the impact of a V-FIA on an ATMEGA328P microcontroller on an Arduino Uno development board, categorizing the faults and their trends concerning the parameters controlled by the attack.A case study to demonstrate the effectiveness of the new FIA method in circumventing access methods.A countermeasure that reduces the attacker’s capabilities to perform an exhaustive search of attack parameters.

The remainder of this paper is organized as follows: Section 2 provides background information to aid in understanding this research. The experiment setup and the performance metrics used in the analysis are explained in Section 3. Section 4 showcases the experiment and its results. Section 5 analyzes and discusses the results. A case study is used to evaluate the performance of the V-FIA proposed under realistic conditions in Section 6. Section 7 proposes a hardware/software countermeasure. Finally, Section 8 concludes the paper and discusses future work.

## 2. Background

In this section, we provide the required FIA information to understand this research. Fault model refers to the set of characteristics that describe the attacker’s capabilities, i.e., a fault model defines the temporal and spatial resolution of the fault injection [25]. Temporal resolution refers to the attacker’s ability to inject a fault at a precise moment, e.g., during the execution of a code section, while spatial resolution is the ability to inject faults in specific places in the circuit, e.g., a data bus or memory. Ultimately, the capabilities required by an attacker to execute a successful attack depend on the target of the attack and the parameters involved.

### 2.1. Fault Injection Attack

Fault injection attacks are generated outside of the operating conditions for which the target devices were designed, such as high temperatures, high or low voltages, and irregular clock cycles. Under such conditions, devices often fail, and the result of the fault is advantageous to the attacker. These attacks can be categorized by the parameters that must be manipulated to cause the fault and by how invasive they are (how much the device must be manipulated to breach it). Multiple and varied are the goals of FIAs, which are used to affect the availability of the target device, the integrity of its information, bypass authentication, and access restricted codes. A distinctive feature of an FIA is the manipulation of parameters to introduce a specific fault; the most commonly attacked parameters are the following:Voltage Glitch: A variation in the operating voltage of the device is generated to change its behavior at a given moment. In this attack, synchronization is essential to meet the required temporal resolution. Voltage faults on the power rails of a smart card are shown in [26]. In addition, the correct control of all the variables involved (e.g., temperature and frequency) must be guaranteed.Clock Glitch: To carry out this attack, the system clock must be manipulated, which can be done by changing the device’s frequency or its duty cycle. In [27], a clock glitch is triggered using the SAKURA-G device to manipulate the clock.Electromagnetic Emission Glitch: The attacked device is subjected to an electromagnetic field that causes the fault that the attacker seeks to exploit. This attack does not require direct manipulation of the target device, so it is one of the most widely used FIAs.Temperature: The temperature can be varied by taking it beyond the specified limits of the device to generate a fault. However, it is more practical to manipulate the temperature to improve the performance of another FIA, as is done in [28]. Some properties of electronic devices change with temperature, such as the impedance, so an increase or decrease in temperature can facilitate the execution of other glitch attacks.Optical Injection: This technique uses lighting transistors in the target device to stimulate conduction and inject the fault into the system. Optical injection with a laser or a high-energy light source such as a UV lamp is executed by decapsulating the chip for precision [29].

However, these attacks are not necessarily limited to manipulating a single parameter. In [30], an FIA is executed combining voltage and heat glitches, while in [28], heat is combined with a clock glitch. Therefore, the parameters that are not part of the attack must be controlled to guarantee the expected result.

The spatial resolution of FIAs is directly related to how invasive the attack is. According to this feature, the FIA can be classified as non-invasive, semi-invasive, or invasive. Non-invasive attacks do not require any type of manipulation of the device; attacks based on electromagnetic pulses or voltage variations belong to this category. On the other hand, a semi-invasive FIA involves mild manipulations of the target device, such as its decapsulation. Finally, invasive FIAs require deep manipulation of the target device, for example, adding new connections or removing parts. The complexity of this analysis often causes the attacked device to become inoperative. More details about FIAs and their main features can be found in [31].

### 2.2. Fault Types and Parameters Involved in FIAs

An FIA can give rise to different behaviors of the target device. It is very difficult to accurately estimate how, when, and where the failure occurred, especially when only the external behavior of the attacked device can be monitored. Some of the typical results of FIAs are device reboot, device disablement, communication problems, modification of the information in memory and registers, and information leak. This research focuses its interest on the modification of information because it can lead to bypassing authentication or accessing prohibited parts of the code. Few investigations perform a detailed analysis of what happens and how it happens in a device to skip an instruction. To the best of the authors’ knowledge, only in [12], has instruction skipping in FIAs been discussed in detail. In this case, fault injection is performed by manipulating the clock cycles for an 8-bit AVR microcontroller. In addition, an electromagnetic FIA is performed for skip instructions, leading to the proposal of an effective software countermeasure in [24]. Countermeasures based on skip instructions are complex and dangerous since they can generate vulnerabilities to other attacks, such as side channel analysis. Finally, in [23], a generic method is proposed to detect FIAs.

The parameters involved in FIAs can be classified into two groups:Parameters of the device under test: resistance, capacitance, inductance, and working frequency.Parameters external to the device under test: temperature, supply voltage, and the size of the voltage interruption window.

We will focus on the external parameters, particularly the supply voltage, interruption window width, and residual voltage, while we keep the other parameters fixed.

## 3. Experiment Setup and Data Analysis

This section describes the proposed fault injection method, the attack model, and the metrics used to analyze the results.

### 3.1. New Fault Injection Method

Voltage fault injections typically involve interrupting the operating voltage or generating a voltage spike. High-precision MOSFETs are usually utilized to interrupt the voltage, resulting in the operating voltage being reduced to zero, or close to zero, for a brief period of time, as described in [11]. This technique necessitates precise control of the time window since, based on tests conducted on an Arduino board’s microcontroller, the interval during which failures can be observed is [8.09, 8.13] μs. No failures are observed below the lower bound, and only device resets are observed above the upper bound. Therefore, this interval must be explored in very small steps of time to obtain an overview of where the exploitable failures occur. This can be good for temporary accuracy, but requires expensive equipment to perform. Another approach is to turn the voltage up, generating a spike for a very short period of time, but this can lead to a higher chance of permanent damage to the devices. In this research, a new method is proposed that uses a low-cost transistor BJT and consists of lowering the voltage close to the lower operating limit specified by the manufacturer of the microcontroller and sustaining it for a period of time in the order of microseconds (the choice of a BJT transistor over a MOSFET is mainly for ease of controlling the current in the base of the transistor and in turn controlling the residual voltage). In this way, a precise device to carry out the attack is no longer needed since there is a much wider pulse width interval to explore the faults, as will be seen later. On the other hand, there is less chance of damaging low-power devices. In Figure 1b, the circuit is presented, inspired by the circuit in Figure 1a presented in [11]. Furthermore, in Figure 2 and Figure 3, the shapes of the window with the conventional method and with the new method are shown. In these images, it can be seen that the interruption generated in the new method has a much higher residual voltage (approximately 1.7 V).

### 3.2. Experiment Setup

To generate the fault injection attack, an ATRIX-7 FPGA on a BASYS3 board was programmed to generate a pulse with a controlled width that activates a switch connected to the output of an HY3005E DC power supply. When the switch receives a pulse, the output voltage is interrupted for a time equal to the width of the pulse. The switch is made up of a 2n222A NPN transistor whose base current is varied to obtain a controlled residual voltage. The device under test is an 8-bit ATMEGA328P-U microcontroller running on an Arduino Uno microcontroller board that is programmed with highly glitch-sensitive code and delivers information through serial communication (all information received by serial communication has a time stamp). This type of code is used [11] because it is easy to inject faults into it and observe them. It forces the processor to continually modify the registers’ information. Since the value of the expected information is known, it is easy to check if the fault changes the register values. In turn, this code also helps in identifying the type of failure occurring in the microcontroller, for example:The appearance of the string “Start” indicates a reset or a jump to the first line of code (depending on the time stamp).Serial buffer compromises are detected when information not present in the code (noise) is displayed through serial communication.It can be inferred that an instruction was skipped or a count register was modified when the value of the variables i, j, k, counter shows an unexpected value.

The code is shown in Algorithm 1. The interval of the interruptions was selected experimentally, as explained in the next section. The parameters measured are the time of the interruption, the operating voltage of the low-power device, the residual voltage, and the temperature. The behavior generated (the fault) by the voltage interruption can be observed and interpreted through the information provided by the microcontroller through serial communication. This information is stored and processed on an external computer. The fault injection attack model can be seen in Figure 4. Voltage interruptions at regular time intervals were generated in the experiment while recording the device’s operation through serial communication. Once a sufficient number of faults were obtained (around $10^{3}$), the attack parameters were changed to collect faults with the new settings. This information was then processed to determine the number of faults of each type depending on the parameters used, allowing for an analysis of trends. Finally, Figure 5 shows the setup used to perform the tests.
**Algorithm** **1:** Sensitive Code
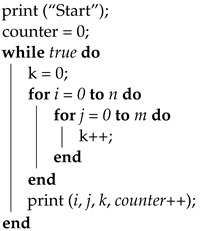


### 3.3. Metrics

The metrics under which the analysis is performed are the following:Failure Rate: failure rate will be understood as the percentage of a certain failure over the total number of failures presented under fixed conditions.Success Rate: the number of attacks necessary to cause any kind of failure.Fault Trends: the fault trend is calculated as the polynomial fit of the failure rate of a given fault over different pulse widths, maintaining all other conditions as constant as possible.

## 4. Experiment and Results

This section explains the parameters used in the experiment, the measurements made, and the results.

### 4.1. Observed Failures and Their Classification

The faults observed during fault injection tests are classified into four distinct groups based on the similarity of their behavior. The first group of faults causes the device to restart, and the second group results in communication errors. The third group of faults leads to changes in the memory register other than the serial buffer, while the fourth group of faults results in the device being inoperable. Each fault’s behavior and the most probable cause of each fault are explained in detail below:Reboot Faults: We have identified three distinct types of reboot faults that can affect the device under test. The first type is characterized by the device becoming unresponsive for an extended period of time, requiring a manual reboot to restore functionality. This behavior may result from the program counter register receiving an invalid value, causing the execution to hang. The second type of reboot fault is indicated by the program starting to execute from the beginning without respecting the designated restart time interval, which can occur if a jump instruction is used to point the program counter to the start of the program. This behavior may be caused by the voltage level of the program counter register being too low, causing it to be interpreted as the first position. The third type of reboot fault occurs when the pulse width is sufficient to cause the device to turn off and on, as if it were being connected and disconnected from a power source. We hypothesize that this behavior corresponds to a blackout reboot, as it conforms to normal reboot times.Serial Communication Faults: This fault can be observed as noise in the communication output as an alteration of a sent value without altering the rest of the values or as the omission of data that should be sent. This fault occurs when the serial communication buffer is affected by the voltage interruption or when the necessary voltage is not reached in the transmission to ensure the completeness of the data. It also may occur because of a problem in the serial receiver of the computer (but this is highly unlikely).Registry Alteration (Glitch): We will call a glitch the faults that change a memory register that can be exploited by an attacker. These can be manifested as an instruction skip or an alteration of a counter in the program, among others. This type of failure can occur for various reasons, such as the alteration of a register that is being actively used in the program or a modification on the program counter that coincides with an instruction stored in program memory, among others.Critical Fault: There is a small probability that the voltage interruption reaches memory spaces reserved to ensure the operation of the device, such as the firmware. By altering these memory registers, the device becomes inoperative and its configuration must be loaded again or permanently ruin the device.

### 4.2. Parameters

In this investigation, we vary three main parameters while maintaining the temperature within a limited range. The parameters being varied are the operating voltage, the residual voltage, and the width of the voltage interruption, with both voltages depending on device features. Meanwhile, we keep the temperature between 19 °C and 23 °C. We vary the operating voltage of the device under test within the parameters recommended by the manufacturer while keeping the residual voltage below the values recommended by the manufacturer. Finally, we determine the range of values for the pulse width experimentally by choosing the interval as follows:The lower limit of the pulse width was selected at which no flaws exploitable by an attacker were observed. Reducing the pulse width to lower than the selected value enters an area where no glitches are observed. This can be seen through the printouts of the fault-sensitive Algorithm 1, where the values i, j, k, counter always have the expected values at the expected time.Similar to the above, the upper limit of the pulse width was determined as the width that no longer generated flaws exploitable by an attacker. Increasing the pulse width more than the selected upper limit causes the device to reset almost every pulse.

Based on the observations, it was decided to carry out the tests within the range of 4.0 to 16.0 μs with steps of 0.5 μs. In this interval, it was possible to observe a considerable number of faults within a reasonable time (a few faults can still be observed under and above the limits, but considerable time is required to obtain a sufficient amount). It is probable that due to the width of the steps when varying the pulse, some fault peaks are lost, which is why the width of the steps is reduced later on in the intervals of interest. On the other hand, the operating voltage and the output voltage were varied in steps of 0.2 V.

### 4.3. Results

In this subsection, we present the outcomes of the study, which will be further explored in the analysis and discussion section. The ATMEGA328P-U microcontroller manufacturer’s recommend an operating voltage of 5 V, with a minimum operating voltage of 1.8 V. Therefore, voltage values were chosen in accordance with these recommendations to conduct the experiments. Based on these recommendations, we chose operating voltage and residual voltage ranges to carry out the tests. The objective is to see how the faults generated by the voltage interruption behave when we drop below the recommended operating voltage and the residual voltage is less than the recommended minimum voltage.

As mentioned before, we classified the faults into four distinct groups. From these, we will first describe the result of critical faults. These critical failures occurred only five times. They occurred in the 12 to 15 μs interval, resulting in three program reloads and two device reboots. Although the number of critical failures identified may not be adequate to provide conclusive evidence, it implies that wider pulse widths could be more susceptible to critical failures.

Next, we present the findings of glitch faults. First, we observed the effect of the operational voltage by varying it while varying the pulse width and maintaining the residual voltage constant, as depicted in Figure 6. The operating voltage was modified in increments of 0.2 V, and the pulse width was varied in increments of 0.5 μs, with a constant residual voltage of 1.7 V. The findings demonstrate that reducing the operating voltage causes the most sensitive area to shift to the left, resulting in more glitches. This is due to the device becoming more unstable and susceptible to failures with fewer disturbances.

Second, the presented data in Figure 7 illustrate the outcome of maintaining the operating voltage at a constant value of 4.8 V while varying the residual voltage. It can be observed that as the residual voltage approaches the recommended voltage determined by the manufacturer, the frequency of failures increases. This finding suggests that operating the device at voltage levels close to the minimum threshold can lead to a greater probability of experiencing malfunctions.

Furthermore, the communication and reboot faults results are presented. Figure 8 and Figure 9 depict the trends of communication and reboot failures, respectively, within the range of interest. It is imperative to note that communication failures are less detrimental to the device compared to reboot failures. Thus, it is advisable to work with pulse widths that minimize the occurrence of reboot failures.

Another interesting result is the trend in the number of trials it takes to obtain a failure of any type as a function of the pulse width. This trend can be seen in Figure 10.

Finally, since the steps in the pulse width are quite large, there is a possibility of losing some important information from the glitches. Therefore, we proceeded to carry out smaller steps of 0.1 μs within the interval of [7.5, 9.5] for the parameters that gave better results (operating voltage of 4.6 and residual voltage of 1.7). The results of this measurement are observed in Figure 11.

## 5. Analysis and Discussion

In this section, we will discuss and analyze the results obtained in order to facilitate their understanding. Firstly, it is important to note that the target device was an ATMEGA328P-U microcontroller integrated into an Arduino Uno development board. It is worth noting that, after conducting attacks on the microcontroller with and without the Arduino board, it was observed that voltage interruptions with greater widths were required when using the Arduino board. This could be attributed to the fact that the components on the board do not discharge instantaneously during voltage interruptions, but with a certain delay similar to a capacitor. This phenomenon causes the voltage to remain higher than the expected residual voltage for a short period of time, providing stability to the device. From this point forward, only the results obtained from attacks on the microcontroller arranged on the Arduino board will be discussed.

The findings depicted in Figure 6 demonstrate that as the device strays from the recommended operating voltage, its instability increases, resulting in a higher frequency of failures. However, when dealing with residual voltage, a contrasting outcome arises. In this regard, it was observed that the closer the device is to the minimum operating voltage but outside the operating range, the more exploitable faults are identified, as illustrated in Figure 7. It was also noted that two interruption width intervals cause failures, namely 7.5–10.5 μs and 12–14.5 μs. However, as Figure 8 and Figure 9 depict, the first interval is prone to communication failures, whereas the second interval is likely to cause reboot failures. Hence, it can be inferred that the first interval not only yields a superior rate of exploitable failures but also has a lower risk of damaging the device. Based on these outcomes, an interval of interest was selected, and tests were conducted with 0.1 μs steps of the pulse width. The results displayed an irregular trend, but a satisfactory percentage of exploitable failures, as demonstrated in Figure 11. Furthermore, Figure 10 reveals that using the optimal parameter settings (operating voltage of 4.6 V, residual voltage of 1.7 V, and a pulse width of 8.5 μs), a failure occurs approximately every two attempts. Thus, it can be deduced that each attempt carries a 50% probability of no effect, a 27.5% chance of a reboot failure, a 20.5% chance of a communication failure, and a 12% chance of a glitch. Finally, it is imperative to consider the temperature during the tests since it has a significant impact on the failure rate [13,28,30]. Despite restricting the temperature variation to between 19 °C and 23 °C, the outcomes might have been marginally affected. Consequently, while attempting to reproduce the experiment, the results may not be identical but should be similar.

## 6. Case Study

The aim of this experiment was to demonstrate a more realistic attack scenario and to validate the effectiveness of the proposed approach. In previous experiments, a code that was susceptible to failures was utilized to achieve the outcomes. In this experiment, we will employ a typical access method that lacks countermeasures and apply the proposed attack method to exploit it. The access technique employed is presented in Algorithm 2, where the code waits for information and compares it with the predefined key. If the key is correct, access to some exclusive part of the code is granted; otherwise, it continues to wait. The same setup and device utilized in Section 3 were used for this experiment, while the parameters were determined based on the optimal outcomes of Section 4.
**Algorithm** **2:** Simple authentication code.
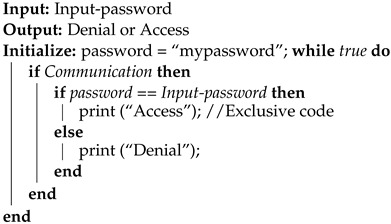


The experiment proceeds as follows: the attack parameters are configured and a voltage interruption is produced every second without any information being transmitted through serial communication; therefore, neither the correct nor incorrect password message should be displayed on the screen. The attack runs for approximately 10 min while the serial communication data are recorded. This process is repeated five times. The results, illustrated in Figure 12, show an average detection of about 50 faults every 10 min.

Before proceeding, it is important to note a few things. First, the faults found in this experiment are not distributed in the same way as those found in the results of Section 4. Second, a fault known as a Print Code has not been previously described, but it is a vulnerability that could be exploited by a malicious attacker. Finally, there are failures that cannot be clearly identified with this implementation, such as when code processing freezes, but these are not the objective of this case study.

The Print Code fault occurs when a sensitive value, such as an authentication key, is stored in an insecure manner, such as being directly assigned to a variable that is compared in the code and saved in flash memory. The behavior of this fault is that the secret key is printed through serial communication without any instruction in the code that explicitly does so.

The Skip Instructions failure occurs when the message “Correct password” is displayed through serial communication, indicating that part of the code was executed without the need to enter any password. Reboot and Communication failures remain the same, with the caveat that some Reboot failures may have been missed and that communication failures are seen as meaningless printouts.

The results of the experiment demonstrate that a simple access method can be bypassed or used to obtain confidential information in less than 1 min with the method used, highlighting the threat that fault injection attacks pose. Even if the key was stored and protected correctly, the skip instruction failure has a high percentage of about 28%, remaining a significant security threat. Finally, this method is a low-cost and low-precision way to observe failures for further study without damaging the device.

## 7. Countermeasure

The proposed countermeasure is designed to mitigate common and less invasive attacks such as voltage and electromagnetic interference FIAs. However, for more invasive attacks such as laser injection or clock injection, additional countermeasures may be necessary. It is challenging to find a countermeasure that is effective against all fault injection attacks, and the countermeasure must be tailored to the specific operating characteristics of the device. It should be noted that implementation of this countermeasure will be part of future work, and this research only describes its general design.

The countermeasure comprises two main components: a hardware part and a software part. The hardware component involves adding a digital potentiometer between the source and capacitor, which is connected to the ground of the device as shown in Figure 13. The software component involves adding code to generate a pseudo-random number that modifies the value of the potentiometer every time the device is turned on or restarted. The capacitor acts as a filter, damping the sudden drop in voltage and discharging on the internal resistance of the device and the configured resistance of the potentiometer. Varying the resistance of the potentiometer makes it difficult to exhaustively search for a pulse width in which exploitable faults are generated. It is crucial to integrate this circuit within the same package to prevent it from being easily physically bypassed.

To select the range of values that the digital potentiometer should take, the internal resistance of the device must be considered. Although the internal resistance of the microcontroller is not provided by the manufacturer, as it depends on various factors such as the operating frequency, temperature, and connected load, an estimated range can be obtained from the voltage and current at which the device operates. For example, the internal resistance range of the ATMEGA328P-U microcontroller is estimated to 20–50 Ω.

The discharge of a capacitor is mathematically described by Equation (Equation 1), which relates to the voltage during the discharge process. In this equation, $V_{F}$ represents the final voltage, $V_{I}$ corresponds to the initial voltage, *R* denotes the discharge resistor, *C* signifies the capacitance of the capacitor, and *t* represents the discharge time. To refine the equation and incorporate additional factors, we introduce two new resistances, namely $R_{i}$ for the internal resistance of the device and $R_{p}$ for the potentiometer resistance ($R_{i}+R_{p}=R$). By substituting these resistances into the equation and solving for time, we derive Equation (Equation 2), which shows that the value of the resistance of the potentiometer is directly related to the discharge time. It is recommended that the resistance of the potentiometer does not exceed the maximum internal resistance of the microcontroller. Additionally, a potentiometer with a resolution that allows the resistance to be varied in small steps is preferred. Several solutions are available on the market that fulfill the resistance requirements.
(1)$$V_{F}=V_{I}\;e^{-\frac{t}{RC}},$$
(2)$$t=-\left(R_{i}+R_{p}\right)\;C\;\ln\frac{V_{F}}{V_{I}}.$$

On the other hand, the capacitance used must be chosen so that the discharge time it adds is significant with respect to the interruption time; in this case, a capacitor between 100 nF and 1 μF is sufficient. It is worth mentioning that a better solution would be to directly use a digitally controllable variable capacitor but we did not find any on the market that met the conditions of acceptable size and capacitance values; due to this, the use of a digital potentiometer was proposed despite the fact that this implies a constant extra energy consumption. Note that the main challenge of this countermeasure from a software point of view is to find a pseudo-random function to generate the random number in a resource-constrained device.

## 8. Conclusions and Future Work

Based on the experiments and the analysis, we can conclude that fault injection attacks are a real threat to low-power devices. The proposed FIA method has proven to be effective in causing different types of failures. In addition, the analysis developed makes it possible to identify its most probable causes. Consequently, the failure rates and trends obtained can be used to optimize the FIA process parameters to achieve the desired failure more accurately. Furthermore, the case study shows that the proposed FIA method can be applied in real scenarios and compromise the security of low-power device authentication methods.

In addition, our findings indicate that devices are more vulnerable to fault injection attacks when operating close to the limit of their specified operating conditions. This highlights the importance of implementing proper countermeasures and testing devices to assess FIA vulnerabilities. The proposed countermeasure presented in this study can serve as a starting point for securing low-power devices against V-FIAs.

In future work, we aim to implement the proposed countermeasure and assess its effectiveness against the proposed fault injection attack method. Furthermore, applying the proposed FIA method to different devices can further test its effectiveness and identify potential vulnerabilities in various low-power devices. Finally, we consider that security mechanisms with collaborative anomaly detection should be investigated, such as methods to detect if nodes of a network are being attacked with fault injections, to thus prevent the attacker from having control of the network.

## Figures and Tables

**Figure 1 sensors-23-07180-f001:**
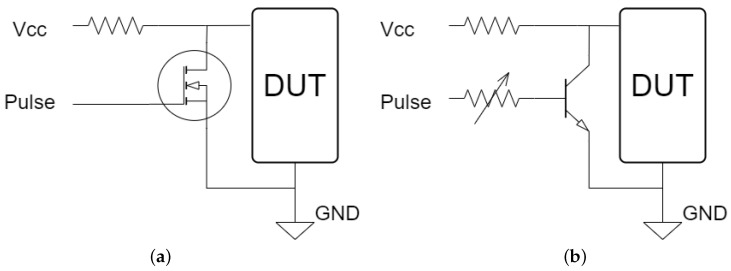
(**a**) Crowbar circuit presented in [11]. (**b**) Circuit topology that allows short-circuiting the operating voltage and controlling the residual voltage magnitude.

**Figure 2 sensors-23-07180-f002:**
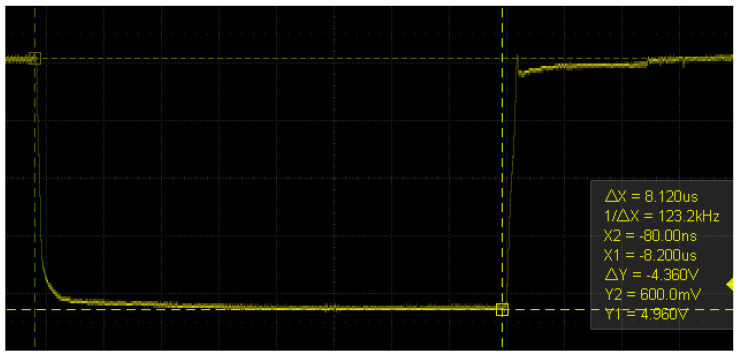
Voltage interruption produced with an IRLML2502 MOSFET and observed with a Siglent SDS 1202X-E oscilloscope.

**Figure 3 sensors-23-07180-f003:**
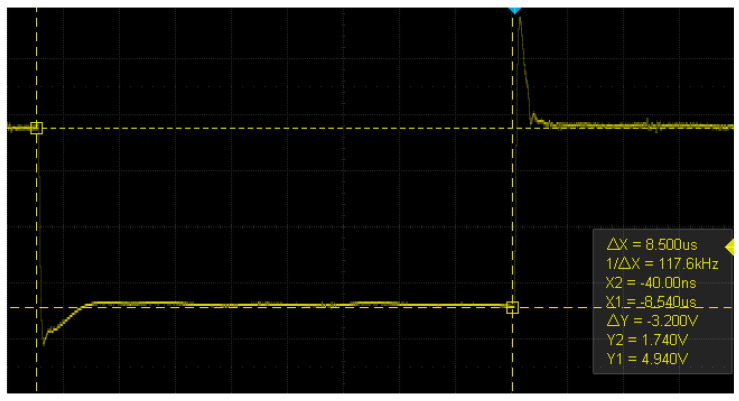
Voltage interruption produced with a 2n222A NPN transistor and observed with a Siglent SDS 1202X-E oscilloscope.

**Figure 4 sensors-23-07180-f004:**
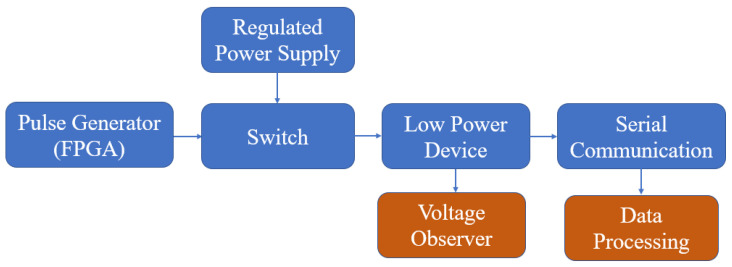
Experiment setup diagram.

**Figure 5 sensors-23-07180-f005:**
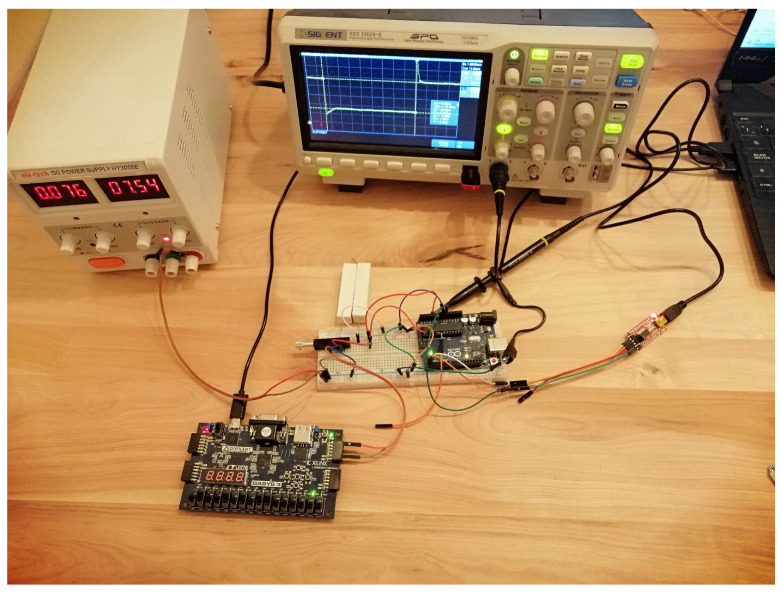
Real setup used in the experiment to carry out the attack using a transistor and regulating the residual voltage.

**Figure 6 sensors-23-07180-f006:**
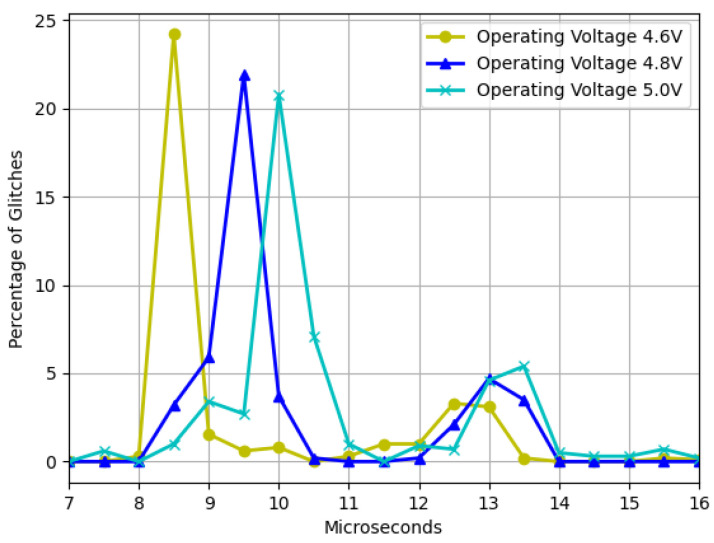
Glitches as a function of the voltage window width with three different operating voltages and a fixed residual voltage of 1.7 V.

**Figure 7 sensors-23-07180-f007:**
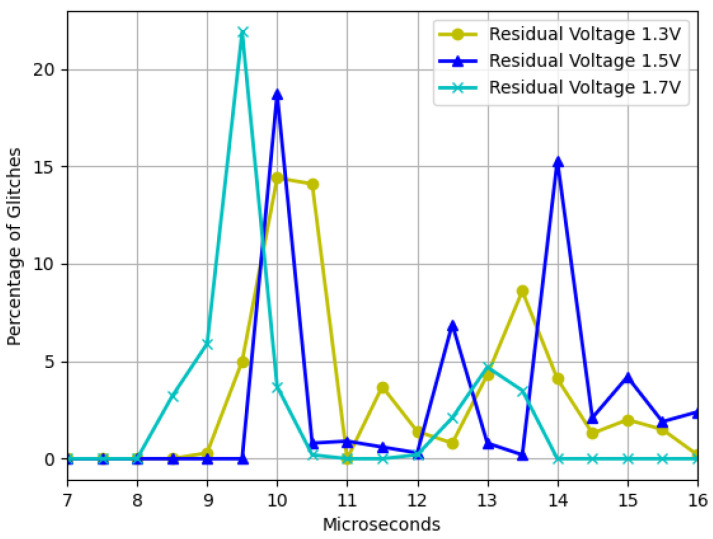
Glitches as a function of the voltage window width with three different residual voltages and a fixed operating voltage of 4.8 V.

**Figure 8 sensors-23-07180-f008:**
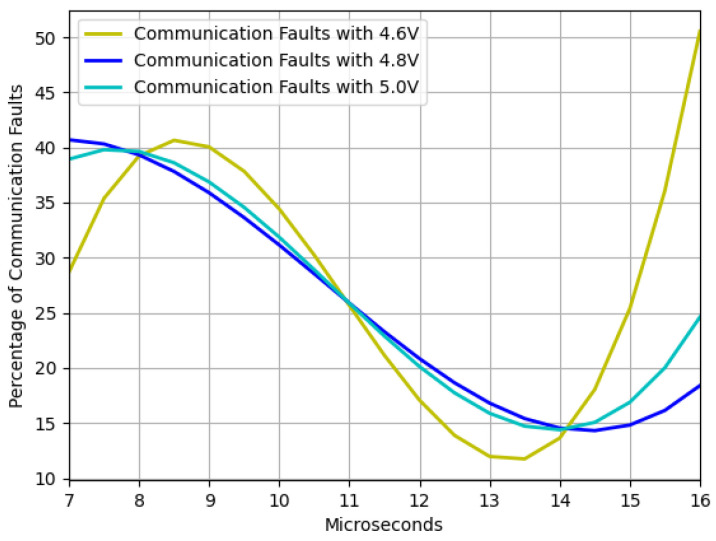
Communication faults trend as function of the voltage window width with three different operating voltages and a fixed residual voltage of 1.7 V.

**Figure 9 sensors-23-07180-f009:**
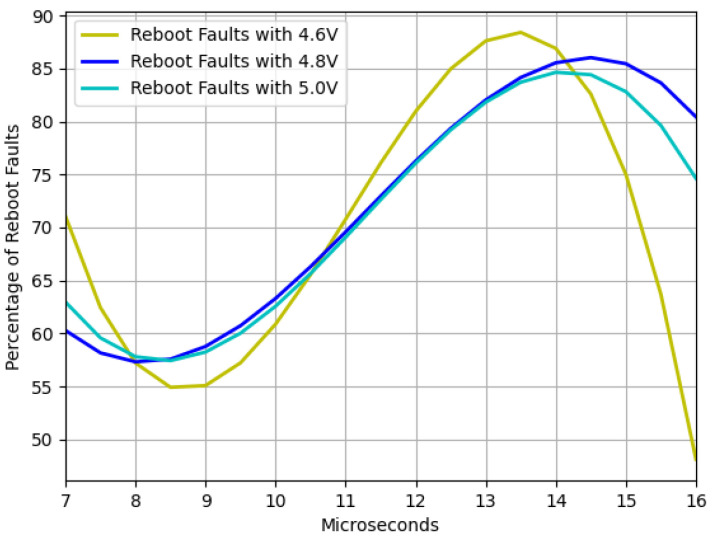
Reboot faults trend as function of the voltage window width with three different operating voltages and a fixed residual voltage of 1.7 V.

**Figure 10 sensors-23-07180-f010:**
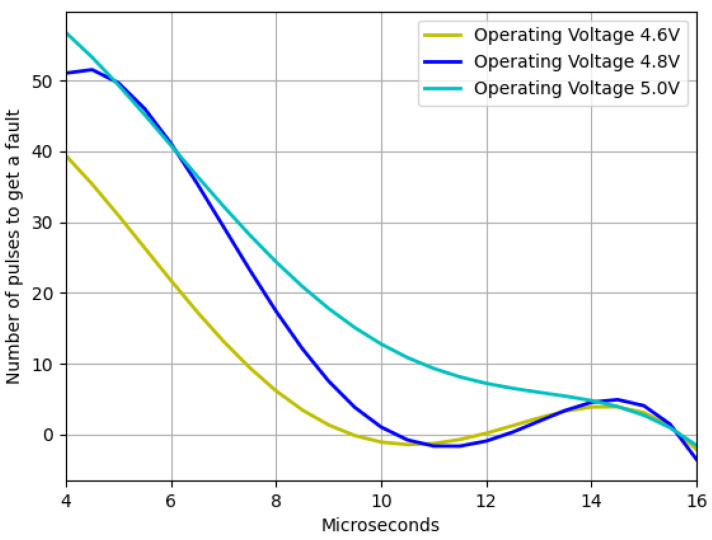
Number of pulses needed to obtain a fault as function of the voltage window width for three different operating voltages and a fixed residual voltage of 1.7 V.

**Figure 11 sensors-23-07180-f011:**
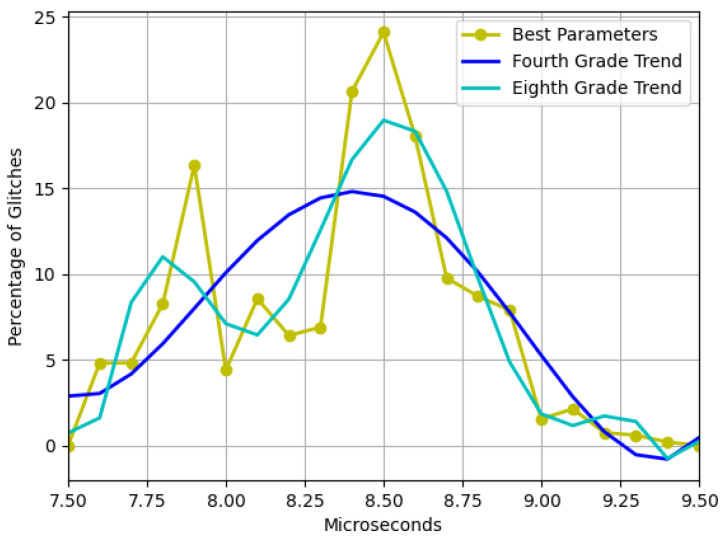
Detailed overview of glitches in the best interval found to attack with the best parameters selected. Trends were found with fourth- and eighth-order polynomial fits.

**Figure 12 sensors-23-07180-f012:**
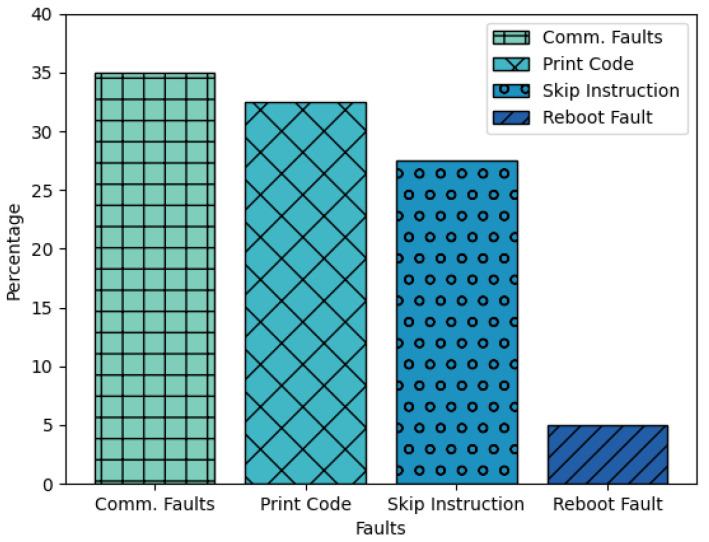
Results of the attack carried out on the device with a simple authentication method.

**Figure 13 sensors-23-07180-f013:**
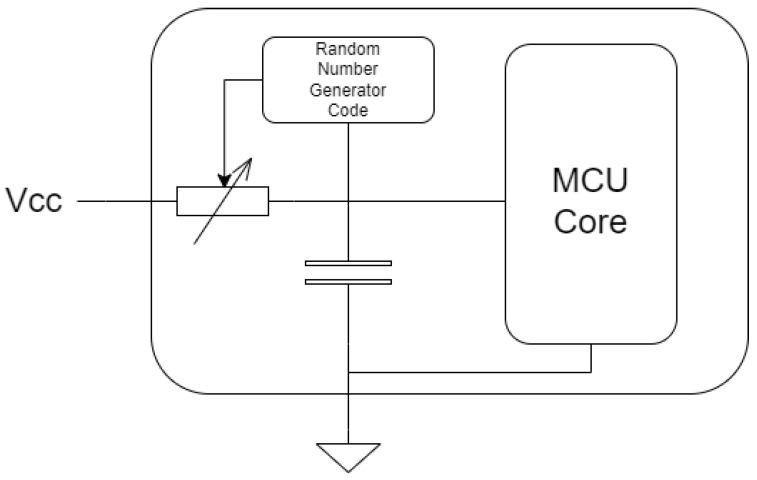
Results of the attack carried out on the device with a simple authentication method.

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
