# Peer review of "A Novel Approach of a Low-Cost Voltage Fault Injection Method for Resource-Constrained IoT Devices: Design and Analysis"

_sensors, 2023, doi:10.3390/s23167180_

Round 1
Reviewer 1 Report
1. The title does not reflect appropriately to the work presented. Better to refine the title.
2. Authors claim that [This analysis not only highlights the feasibility of a V-FIA but also describes the main characteristics of the resulting failures and how they propagate in a typical resource-constrained device. Finally, we designed a hardware/software countermeasure that can be integrated into any resource-constrained device to prevent such attacks in IoT scenarios.]. Better to provide any benchmarking you can find, such as for the mentioned papers which consider the Encryption Selection Model for IoT Devices Based on IoT Device Design (10.23919/ICACT53585.2022.9728914) and (10.23919/ICACT53585.2022.9728960) where authors presented IoTES (A Machine learning model) Design dependent encryption selection for IoT devices.
3. Authors claim in the introduction for the paper contribution is that [A new approach to control the outcome of a fault injection attack through monitoring residual voltage in the voltage interruption window.]. while in the title and abstract they mention that their research work is providing the analysis not any new approach. Authors need to be pretty clear through out the paper, what they are proposing.
4. Algorithm 1 is a simple description of a simple loop functionality, not the algorithm, authors require to provide the valuable Algorithm,
5. Results require more elaboration for the better understanding of the readers.
6. Conclusion needs to be mapped with the abstract properly, for the claims made and what results achieved.
7. Proofread is recommended for the entire paper.
Please refer to the set of comments.
Reviewer 2 Report
In this submission, the authors proposed a variation of the conventional V-FIA approach and utilize it to attack an 8-bit AVR microcontroller. They determined the optimal attack parameters and utilized them in a case study where they applied the new attack technique against an authentication method, thus demonstrating its effectiveness in a real-world scenario. They investigated both the feasibility of a V-FIA, as well as describing the main characteristics of the resulting failures and how they propagate in a typical resource-constrained device. The authors also designed a hardware/software countermeasure that can be integrated into any resource-constrained device.
I find this manuscript to be of interest to Fault, security, Sensors and IOT researchers as well as readers of this journal. As such, I am generally supportive of publication with a few minor notes that should be incorporated. While the authors describe their work in the context of Fault, security, sensors and IOT, there has been much prior work on Fault, security, sensors and IOT which should be noted:
Nazari Cheraghlou, M., Khadem-Zadeh, A., & Haghparast, M. (2019). A new hybrid fault tolerance approach for Internet of Things. electronics, 8(5), 518.
Nazari Cheraghlou, M., Khadem-Zadeh, A. & Haghparast, M. A Novel Hybrid Fault Tolerance Architecture in the Internet of Things. Wireless Pers Commun 118, 383–411 (2021). https://doi.org/10.1007/s11277-020-08019-1
In particular, these prior studies determine how to utilize hybrid fault tolerance approach for Internet of Things and also determine a framework for optimal fault tolerance protocol selection on IoT sensor layer and produce solutions which should be noted. With this minor edit, I would be willing to re-review this manuscript for subsequent publication in Sensors Journal.
Reviewer 3 Report
This manuscript is organized, concise, and well-written in general. The introduction is relevant and theory-based. Enough information about the results of studies is provided for readers to understand the current study's rationale and procedures. The methods are generally appropriate.
Highlight the importance of the proposed work in comparison with the existing techniques in the literature review. Each paper should make it clear in the literature review what the proposed methodology, novelty, and experimental results are. At the end of related works, highlight better in some lines what overall technical gaps were observed in existing works that led to the design of the proposed approach. To better delineate the context and the different possible solutions, you can consider the following papers as references: https://www.sciencedirect.com/science/article/abs/pii/S002002551630737X and https://link.springer.com/article/10.1007/s11042-017-4443-1.
The depth and breadth of the literature survey are not enough, and it is necessary to supplement the advantages and weaknesses of directly related research on the basis of an extensive literature review.
Reviewer 4 Report
The authors introduced a new fault injection attack by monitoring residual voltage in the voltage interruption window and showed the effectiveness through experiments. The hardware/software countermeasures for fault injection attacks are proposed. If the author can show some experimental results for the countermeasure that will be great.
Round 2
Reviewer 1 Report
Authors addressed all the comments and concerns, the manuscript stands for acceptance.
Minor edits are recommended.
Reviewer 2 Report
Congratulations on the good work.